# Estimating the Contribution of Renal Function to Endothelial Dysfunction and Subclinical Inflammation with a Two-Cohort Study: Living Kidney Donors and Their Transplant Recipients

**DOI:** 10.3390/ijms26199535

**Published:** 2025-09-29

**Authors:** Irina B. Torres, Carla Burballa, José M. González-Posada, Domingo Hernández, Esteban Porrini, Janire Perurena, Vicente Cortina, Manel Perelló, Dolores Redondo-Pachón, Ana González-Rine, Mercedes Cabello, Maria José Pérez-Sáez, Marta Crespo, Oriol Bestard, Daniel Serón, Francesc Moreso

**Affiliations:** 1Nephrology Department, Hospital Universitari Vall d’Hebron, 08035 Barcelona, Spain; irinabetsabe.torres@vallhebron.cat (I.B.T.); manel.perello@vallhebron.cat (M.P.); oriol.bestard@vallhebron.cat (O.B.); danielseron56@gamil.com (D.S.); 2Department of Medicine, Universitat Autònoma Barcelona, 08035 Barcelona, Spain; 3Nephrology Department, Hospital del Mar-Parc de Salut Mar, 08003 Barcelona, Spain; cburballa@psmar.cat (C.B.); mredondopachon@psmar.cat (D.R.-P.); mperezsaez@psmar.cat (M.J.P.-S.); mcrespo@psmar.cat (M.C.); 4Nephrology Department, Hospital Universitario de Canarias, 38320 La Laguna-Tenerife, Spain; jmgposada@hotmail.com (J.M.G.-P.); rinneanag@yahoo.es (A.G.-R.); 5Nephrology Department, Hospital Regional Universitario Carlos Haya, 29010 Málaga, Spain; domingohernandez@gmail.com (D.H.); mcabello82@hotmail.com (M.C.); 6Instituto de Tecnologías Biomédicas, University of La Laguna, 38320 La Laguna-Tenerife, Spain; esteban.l.porrini@gmail.com; 7Immunology Department, Hospital Universitari Vall d’Hebron, 08035 Barcelona, Spain; janine.perurena@vallhebron.cat; 8Hematology Department, Hospital Universitari Vall d’Hebron, 08035 Barcelona, Spain; vicente.cortina@vallhebron.cat

**Keywords:** kidney transplantation, living donor, renal function, iohexol clearance, endothelial dysfunction, inflammation

## Abstract

Living kidney transplantation offers the best results for end-stage renal disease patients, but concerns about cardiovascular risk after nephrectomy for kidney donors have been raised. We aimed to estimate the contribution of renal function to endothelial dysfunction (ED) and subclinical inflammation in a non-interventional, prospective, multicenter, longitudinal study with two cohorts: living kidney donors and their transplant recipients (registered clinical trial NCT02515643). The measured glomerular filtration rate (mGFR) by iohexol clearance, estimated GFR according to the CKD-EPI and MDRD-4 formulas, and levels of endothelial dysfunction (sVCAM-1, sICAM-1, E-selectin, von Willebrand Factor, pentraxin, and urinary albumin-to-creatinine ratio) and subclinical inflammation biomarkers (sIL-6, sTNF-R1, sTNF-R2, sTWEAK, and high-sensitivity C-reactive protein) were determined at baseline and 1-year follow-up. Fifty pairs of donors and recipients were recruited between 2015 and 2018. Among the endothelial dysfunction biomarkers, sVCAM-1 increased in donors and decreased in recipients (*p* < 0.01) while, among the inflammation biomarkers, sTNFR1 and sTNFR2 significantly increased in donors and decreased in recipients (*p* < 0.001). After transplantation, parallel increases and decreases in ED and subclinical inflammation biomarkers were observed in the donor and recipient cohorts, respectively. Long-term follow-up is needed to characterize the cardiovascular risk associated with these changes.

## 1. Introduction

In recent years, chronic kidney disease (CKD) has become an increasing global health issue, and even mild reductions in the glomerular filtration rate (GFR) have been associated with increased cardiovascular (CV) risk, a condition that is much more prevalent than the progression to end-stage renal disease (ESRD) [1]. Although the K-DIGO guidelines indicate that CV risk increases when the GFR is lower than 60 mL/ min/1.73 m^2^ [2], some studies suggest that such risk occurs on a continuum and that lower reductions in GFR are also associated with increased CV risk [3,4,5]. Similarly, the presence of albuminuria has been associated with an increased CV risk and, in the general population, low levels of albuminuria—even below the normal range (<30 mg/day)—have been independently associated with higher CV risk [6]. Moreover, the effects of microalbuminuria and the reduction in GFR are independent and have additive effects. The abovementioned studies suggest an association between CKD and CV risk, but a causal association has not been clearly demonstrated. Cross-sectional studies performed in patients with CKD present confounding variables, such as hypertension, diabetes and dyslipidemia, and other comorbidities associated with CKD (e.g., inflammation, anemia). Therefore, it could be argued that in this situation, the high CV risk is related to several CV risk factors with additive effects, and that CKD per se does not modify the CV risk. To establish causality between the two processes of CKD and CV risk, a longitudinal study design will be necessary to permit observation of the progressive reduction in GFR and/or the appearance of microalbuminuria over time. In this regard, living kidney donation offers a unique opportunity to establish a quasi-experimental model involving healthy individuals who suffer from a 50% abrupt reduction in GFR that prompts adaptive mechanisms in response to renal mass reduction [7,8,9], while approximately 50% of GFR is restored in patients with end-stage renal disease who have received a kidney from their living donor.

Kidney transplantation (KT) is the treatment of choice for ESRD since it restores renal function and is associated with longer patient survival and better quality of life [10]. Although KT is associated with a lower mortality rate than other renal replacement therapies, the standardized mortality rate among patients who have undergone KT remains too high compared to the general population [11]. In other countries, like USA and northern Europe, living kidney transplantation represents up to 50% of all transplants performed. In Spain, as an attempt to address the shortage of kidneys for transplantation, living donation has been expanded from the beginning of the current century. During the first years of this expansion, the selection criteria for donors were very strict, being limited to a selected population with a very low CV risk; however, in recent years, the criteria have changed and older donors (up to 70 years old) with more comorbidities (well-controlled hypertension or overweight) are also being accepted. However, this subgroup of living donors under the “expanded criteria” may face an increased CV risk after donation. Nephrectomy for kidney donation has been associated with different modifications in kidney function, proteinuria, blood pressure, glucose metabolism, and pregnancy outcomes, as well as lipid and mineral metabolism disturbances [12,13,14,15,16,17]. However, research on the effects of nephrectomy on patient survival and CV risk has yielded contradictory results. According to a study conducted in the USA that included more than 80,000 donors who were followed for 6 years, mortality was slightly inferior when compared to control subjects of the NHANES study [18]. Another study conducted in Canada that included more than 2000 donors and compared them to selected healthy individuals (n = 20,000) showed that donors had lower mortality and less CV events [19]. However, a Norwegian study comparing living donors with potential healthy donors showed that CV mortality increased in patients who donated their kidney, especially donors older than 60 years [20,21]. Importantly, the survival curves of living donors and healthy individuals did not diverge until 10 years after donation [20]. Recently, in a study with the same Norwegian living donor cohort, an increased risk of ischemic heart disease was observed compared with healthy controls during a follow-up period longer than 10 years (adjusted odds ratio = 1.64) [22].

Mechanisms that increase CV risk in CKD have been widely studied, and several studies have shown that endothelial dysfunction (ED) and chronic inflammation are associated with CKD [23]. The gold standard for measuring ED is to measure the changes in brachial blood flow after the infusion of intra-arterial agents as being dependent or independent of the endothelium [24]. Because this is an invasive procedure, research in this area has focused on investigating biomarkers of ED in both blood (e.g., soluble factors like vascular cell adhesion molecule or von Willebrand factor) and urine (microalbuminuria) [25]. Inflammation, as a complex biologic response of vascular tissue to different injuries, enables the removal of the inciting agent and initiates the healing process. However, impaired excretory renal function prolongs the plasma half-life of several proinflammatory cytokines, which may result in an enhanced inflammatory load. A variety of cytokines and acute-phase proteins are released to regulate the inflammatory response. However, unregulated chronic systemic inflammation may contribute to an increased risk for clinical and subclinical CV diseases, leading to increased morbidity and mortality in CKD patients [26].

In the present study, we aimed to estimate the contribution of renal dysfunction to ED and subclinical inflammation biomarkers in two patient cohorts—living kidney donors and their transplant recipients.

## 2. Results

### 2.1. Patient Characteristics

Between July 2015 and February 2018, 53 donor–recipient pairs were recruited. One donor–recipient pair refused to sign the consent form before the initiation of study procedures and two recipients suffered from graft thrombosis during the initial 24 h after transplant. Thus, 50 pairs were included in the study (the CONSORT diagram is shown in Figure 1).

The demographics, anthropometric data, and comorbidities of the 50 donor–recipient pairs included in the study are shown in Table 1. Prediabetes was diagnosed in five donors at baseline (impaired glucose tolerance (*n* = 2) and impaired fasting glucose (*n* = 3)) and in six cases at 1-year follow-up (impaired glucose tolerance (*n* = 6)). Glycosylated hemoglobin (5.4 ± 0.2 vs. 5.4 ± 0.2%) and HOMA-IR (2.53 ± 2.04 vs. 2.59 ± 2.01 mU/mL/mmol) did not change from baseline to 1-year follow-up in the donor cohort. There were no significant changes in lipid metabolism after nephrectomy in the donor cohort. Neither the donors nor recipients had cardiovascular events before transplantation.

Regarding transplant-related variables in the recipient cohort, HLA mismatches were 1.13 ± 0.62, 1.28 ± 0.62 and 1.02 ± 0.65 for the A, B and DR loci, respectively. All kidney transplants displayed immediate kidney function after transplantation, and biopsy-proven acute rejection during the first year was diagnosed in six cases (12%). Rejection episodes were classified according to the Banff criteria as T-cell-mediated rejection (TCMR) grade IA (*n* = 3), grade IB (*n* = 1) and grade IIA (*n* = 1) and active antibody-mediated rejection (*n* = 1). During follow-up this last patient lost his graft due to chronic active antibody-mediated rejection at 7 months. Post-transplant diabetes mellitus was diagnosed in six cases (12%). No major surgery-related complications were recorded in the pool of donors. Body weight did not significantly change at one year in the donor cohort (72 ± 13 kg vs. 73 ± 14 kg) and, as expected, increased significantly in the recipient cohort (74 ± 16 vs. 79 ± 16 kg, *p* = 0.002).

### 2.2. Renal Function at One Year

Serum creatinine, mGFR before nephrectomy in the donors and at one year in both the donors and recipients, and eGFR-CKD-EPI and eGFR-MDRD-4 at baseline and one year are shown in Table 2.

The Bland–Altman analysis using mGFR as the reference method showed that the bias of baseline eGFR-CKD-EPI and eGFR-MDRD-4 was low (−6.2 ± 14.9 and −2.3 ± 17.5 mL/min/1.73 m^2^, respectively), but the absolute percentage error was significant (11.2% and 12.1%, respectively). Additionally, there was a proportional bias since both formulas overestimated for lower values and underestimated for higher values (β = −0.60 and *p*-value < 0.001 for eGFR-CKD-EPI and β = −0.59 and *p*-value < 0.001 for eGFR-MDRD-4; Figure 2). Similarly, in the donor cohort, the bias was low at 1-year follow-up (0.89 ± 14.3 for eGFR-CKD-EPI and −3.9 ± 14.5 mL/min/1.73 m^2^ for eGFR-MDRD-4), but the absolute percentage error was significant (10.4% and 16.6%, respectively); once again, there was a proportional bias since both formulas overestimated for lower values and underestimated for higher values (β = −0.75 and *p*-value < 0.001 for eGFR-CKD-EPI and β = −0.86 and *p*-value < 0.001 for eGFR-MDRD-4; Figure 2). As expected, donor mGFR was negatively associated with donor age (R = −0.43, *p*-value = 0.02).

Finally, in the case of recipients, the one-year bias was 7.2 ± 14.9 for eGFR-CKD-EPI and −0.6 ± 11.1 mL/min/1.73 m^2^ for eGFR-MDRD-4, with an absolute percentage error of 22.5% and 13.2%, respectively. In the case of recipients, a proportional bias was not found.

### 2.3. Endothelial Dysfunction and Inflammation Biomarkers

The levels of endothelial dysfunction and inflammation biomarkers in the donors and recipients at baseline and 1-year follow-up are shown in Table 3.

Among the endothelial dysfunction biomarkers measured in this study, sVCAM-1 increased in the donors (from 639 ± 197 to 718 ± 213 ng/mL; *p*-value = 0.0027) and decreased in the recipients (from 1208 ± 417 to 943 ± 313 ng/mL; *p*-value < 0.001) while sICAM-1, PECAM-1, vWF and PTX-3 did not experience significant changes in the donor cohort. UACR did not change significantly at 1 year follow up in the donor cohort (from 5.9 ± 6.3 to 10.5 ± 24.0 mg/g; *p*-value = 0.217). Remarkably, one donor showed an increase in UACR from 29 mg/g at baseline to 124 mg/g at 1-year follow up (a 64-year-old female, who was overweighted with a BMI of 28.4 kg/m2 and had hypertension that required taking one drug per day, with a non-dipper pattern in the ABPM and with impaired glucose tolerance in the OGTT). In the other cases, UACR was lower than 30 mg/g at baseline and at 1-year follow up.

In the recipient cohort, E-selectin (from 37 ± 19 to 34 ± 14 ng/mL; *p*-value = 0.0225) and PECAM-1 (from 79 ± 23 to 71 ± 15 ng/mL; *p*-value = 0.0016) decreased after transplantation while the reduction in vWF approached significance (from 181 ± 53 to 167 ± 57%; *p*-value = 0.0694).

Among the inflammation biomarkers, sTNFR1 and sTNFR2 significantly increased in the donors and decreased in the recipients (*p* < 0.001 for both comparisons), while non-significant differences were observed in sIL-6, sTWEAK and hs-CRP (Table 3).

### 2.4. Relationship Between Renal Function and Biomarkers

At baseline, there was no correlation between mGFR and ED markers (sVCAM-1, sICAM-1, E-selectine, sPECAM-1, vWF and PTX-3) in the donor cohort, but there was a moderate correlation between mGFR and UACR (R = 0.49, *p* = 0.003). Subclinical inflammation biomarkers (sIL-6, sTNF-R1, sTNF-R2, sTWEAK and hsCRP) were also not correlated with mGFR at baseline in the donor cohort.

At 1-year follow up, there was no correlation between mGFR and ED biomarkers (sVCAM-1, sICAM-1, PECAM-1, vWF, PTX-3 and UACR) in the donor cohort. In the recipient cohort, there were weak correlations between mGFR and sVCAM-1 (R = −0.36, *p* = 0.041) and mGFR and sICAM-1 (R = −0.35, *p* = 0.044).

Among the subclinical inflammation biomarkers, sTNF-R1 was moderately correlated with mGFR at 1-year, in both the donor (R = −0.40, *p* = 0.016) and the recipient cohorts (R = −0.55, *p* = 0.001) (Figure 3). Similar results were obtained for sTNF-R2. Finally, sIL-6, sTWEAK and hsCRP did not correlate with mGFR at 1-year follow up neither the donor nor the recipient cohort.

### 2.5. Blood Pressure and Atherosclerotic Burden in the Donor Cohort

Office blood pressure and ABPM at baseline and at 1-year follow up in the donor cohort are shown in Table 4. Eight living donors had well-controlled hypertension before kidney donation. Blood pressure measured via ABPM did not show significant changes during the first year after kidney donation but the proportion of non-dipper patients slightly increased (from 18 to 34%).

The median number of carotid plaques at baseline was 0 (interquartile range [IR]: 0–2), the mean carotid IMT was 0.72 ± 0.25 mm, the mean PWV was 7.6 ± 1.6 m/s and the mean ABI was 1.19 ± 0.19. The number of carotid plaques was correlated with carotid IMT (rho = 0.47; *p* = 0.026) and ABI (rho = −0.72; *p* < 0.001) but not with PWV (*p* = 0.932).

Daily mean blood pressure monitored by ABPM was not associated with patient’s age, gender, BMI, smoking habits or surrogates of atherosclerotic burden in the donor cohort. Surrogates of atherosclerotic burden at baseline in the donor cohort were associated with patient’s age (PWV, R = 0.76, *p* < 0.001; carotid plaques, rho = 0.37, *p* = 0.032; carotid IMT, R = 0.54, *p* = 0.001) and with mGFR (PWV, R = −0.60, *p* = 0.003; carotid plaques, rho = −0.34, *p* = 0.049 and carotid IMT, R = −0.42, *p* = 0.046) but not with patient’s gender, BMI or smoking habits. In the multivariate analysis, patient’s age was the only independent predictor of atherosclerotic burden, while mGFR was not included in the models.

### 2.6. Follow-Up at 5 Years

The donors and recipients were regularly followed up at the participating centers. At the end of the follow-up period, all but one of the donors were alive, and none presented a CVE. One donor died at 9 years due to metastatic colon cancer. Fifteen donors (30%) had well-controlled hypertension with a mean number of 1.3 ± 0.6 antihypertensive drugs. Arterial hypertension at five years was associated with age (58 ± 10 vs. 50 ± 12 years, *p* = 0.0385) and with eGFR at 5 years (58 ± 15 vs. 69 ± 10 years, *p* = 0.007) but not with VCAM-1 levels or sTNFR1/sTNFR2. In the multivariate analysis, eGFR was the only variable associated with arterial hypertension (odds ratio: 0.93, 95% confidence interval 0.87–0.99, *p*-value = 0.029). Additionally, one donor developed diabetes mellitus at 3 years under treatment with antidiabetic oral drugs and associated with microalbuminuria. One donor presented obstructive acute kidney failure at one year due to urinary lithiasis, but recovered after placement of a ureteral stent.

In the recipient cohort, three patients died with a functioning graft (lung carcinoma at 50 months, neurodegenerative disorder with broncho-aspiration at 62 months, metastatic melanoma at 97 months) and five grafts failed (chronic antibody-mediated rejection at 7 months, antibody-mediated in a non-adherent to immunosuppressive treatment patient at 48 months, recurrent amyloidosis at 68 months, donor-related nephroangiosclerosis at 74 months and microvascular inflammation, negative for HLA donor-specific antibodies and C4d at 94 months). Thus, patient survival was 94% and death-censored graft survival was 90%. After transplantation only three patients had a non-fatal CVE (one cerebrovascular accident and two coronary ischemic events requiring revascularization).

We analyzed evolution of renal function in the donor and recipient cohort from baseline to 5 years. In the donor cohort, renal function decreased after donation, and a progressive increase was observed until 5 years, while in the recipient cohort renal function improved after transplantation and slightly decreased until 5 years (Figure 4, left panel). From the first month until 5 years the linear mixed-effects model with random intercept by patient showed a positive slope for eGFR in the donor group (+0.10 mL/min/1.73 m^2^ per month; *p* = 0.012) and a slightly negative slope in the recipient group (−0.03 mL/min/1.73 m^2^ per month), with a significant difference between slopes (−0.13 mL/min/1.73 m^2^ per month; *p* = 0.019) (Figure 4, right panel).

As expected, there were significant discrepancies between donors and recipients during follow-up, as depicted in the Figure 5.

## 3. Discussion

In the present study, we evaluated renal function in living donors and their renal transplant recipients from the date of transplantation until five years after. We observed that renal function in the living donors is overestimated when using standard formulas, especially in recipients with a lower measured GFR by iohexol clearance. Renal function diverged from one month onwards, showing a significant improvement in the donor cohort and a slight decrease in the recipient cohort. Regarding biomarkers, we observed an increase in sVCAM-1 in the donor cohort at 1-year follow-up that did not correlate with renal function. Additionally, sTNFR1/sTNFR2 increased in the donor cohort and this change was moderately related to the decrease in renal function. As expected, in the recipient cohort, we observed a significant improvement in some endothelial dysfunction biomarkers (sVCAM-1, E-selectin, and PECAM) and inflammation biomarkers (sTNFR-1 and sTNFR-2), which displayed low to moderate correlations with renal function at 1-year follow-up.

Living donor kidney transplantation is the best kidney replacement therapy option for eligible patients with ESRD, offering superior outcomes compared with deceased donor transplantation. Recognition of its benefits to recipients and society has led to efforts to promote living donations by different scientific societies [27]. Simultaneously, the success of living donation programs depends on the capability to ensure the safety of and good outcomes in living kidney donors, which relies on an in-depth evaluation and careful risk assessment before donation. The workup for living donation includes the evaluation of individual variables related to current kidney health (GFR, proteinuria or albuminuria, and hematuria) and the assessment of metabolic and cardiovascular risk factors, such as hypertension, impaired glucose tolerance, obesity, smoking, and genetic risk factors such as family history of diabetes [28]. It is evident that assessment of kidney function is crucial to the donor candidate evaluation process, as donor nephrectomy is followed by adaptive hyperfiltration to achieve approximately 60–70% of pre-donation kidney function. In fact, this represents one of the relatively few scenarios in nephrology where an accurate assessment of GFR is essential. Many guidelines are available (reviewed in [28]), but they vary in regard to their recommendation on the use of methods, and no guideline provides details on the choice of exogenous marker or the choice of protocol. Notably, the widely accepted threshold of 80 mL/min/1.73 m^2^ should be adopted while taking into consideration the donor’s age, as recommended in different guidelines [28]. In our study, acceptance of living donors relied on eGFR-CKD-EPI, 24-h creatinine clearance, and/or isotopic mGFR, depending on the policy and available methods at each participating center. Unfortunately, GFR as measured by iohexol plasma clearance was only available at the end of the study and, thus, was not employed when making clinical decisions. Importantly, in our sample, we observed that there were 9 patients out of 50 (18%) with eGFR-CKD-EPI higher than 80 mL/min/1.73 m^2^ but with a measured GFR below 80 mL/min/1.73 m^2^. These data confirmed the findings of previous studies showing that eGFR is not reliable for the evaluation of renal function in candidates for living kidney donation and may lead to the acceptance of candidates with reduced renal function in more than 10% of cases [29]. The long-term consequences for living kidney donors deserve further studies with longer follow-up.

A link between GFR reduction and CV risk in living donors has not been properly established, and contradictory results have been obtained from different registries [18,19,20,21,22]. It is important to remark that the available studies suggest that CV risk is not significantly increased during the first 10 years after donation, though registry studies with longer follow-up observed an increased CV risk. Thus, early determination of surrogates of ED and subclinical inflammation can contribute to characterizing CV risk in living donors. Laboratory measurements of plasma levels for different molecules and/or other elements could be useful to assess endothelial damage and activation. Considering that there is no universal marker of endothelial damage, we have evaluated different molecules. A study conducted in patients with advanced chronic renal failure has shown that CKD (vs. healthy controls) is associated with high levels of sVCAM-1, sICAM-1, and vWF, but not with sPECAM-1 or E-selectin [30]. We added to these biomarkers the determination of PTX3, a multifunctional protein identified as a cognate molecule of C-reactive protein that has complex regulatory roles in inflammation and extracellular matrix organization and remodeling [31]. In our study, we observed a reduction in ED biomarkers—sVCAM-1, sPECAM-1, E-selectin, and vWF—in the renal transplant cohort, while sICAM-1 and PTX-3 did not experience significant changes. This observation agrees with another study comparing different biomarkers of ED and subclinical inflammation in patients with non-dialysis ESRD, patients with ESRD on dialysis and renal transplant recipients [32]. In that study, a small cohort of 15 kidney transplant recipients were followed for six months and sVCAM-1, but not sICAM-1, significantly decreased after transplantation. Thus, our study confirms that restoration of renal function after transplantation reduces the levels of certain biomarkers of ED in renal transplant recipients. Conversely, in the donor cohort, we only observed a significant increase in sVCAM-1, while the levels of the other biomarkers tended to remain stable. Levels of sVCAM-1 have been proposed as predictors of mortality and morbidity in patients with chronic heart failure and endothelial injury in patients with coronary artery disease [33]. Since sVCAM-1 is up-regulated in different cardiovascular diseases, the possible contribution of this adhesion molecule to the pathogenesis and development of CVD has been debated in the literature (reviewed in [33]). In a cohort study of patients with a wide range of CKD (mean eGFR of 43 ± 20 mL/min/1.73 m^2^), the serum levels of sVCAM-1 were correlated with renal function and ED measured by flow-mediated dilation (FMD) after adjustment for different CKD risk factors and the use of different medications, supporting that ED is present in established CKD [34]. In this cohort, E-selectin and vWF were also associated with CKD, while sICAM-1 levels were not significantly different between CKD patients and healthy controls. Notably, in a study conducted in Turkey with a cohort of 32 living donors, it was also observed that VCAM-1 increased 1 year after donation (from 680 ug/mL at baseline to 961 ug/mL at 1 year), and this increase was moderately correlated (r = −0.42) with ischemia-induced flow-mediated dilation (FMD) of the brachial artery [35]. Moreover, in the multivariate analysis, VCAM-1, serum uric acid, and eGFR were independent predictors of FMD 12 months after kidney donation. Taking together, these results suggest that sVCAM-1 is a more sensitive ED biomarker in patients with mild CKD like living kidney donors. Even though statins have been shown to significantly lower sVCAM-1 and sICAM-1 [36], the low number of patients treated with statins in our living donor cohort (*n* = 5) precluded further analysis.

We also explored the contribution of renal function to subclinical inflammation by monitoring some well-known biomarkers for CKD. In our study, hsCRP and sIL-6 tended to slightly increase in the living donors and decrease in the transplant recipients, but the wide ranges of observations in both cohorts precluded the detection of significant changes. Similar results for these two biomarkers were observed in the abovementioned study after renal transplantation [32], suggesting that renal function is not associated with these biomarkers. Additionally, both studies suggest that pre-emptive renal transplantation (60% of patients in our sample) is associated with a lower inflammation environment compared with dialysis.

Importantly, a highly significant change was observed for sTNFR1 and sTNFR2 in both cohorts. TNF receptor-1 (TNFR-1) is a cell surface receptor expressed in the capillary endothelium that plays a causative role in the development of endothelial cell dysfunction and inflammation [37]. In recent years, an association of serum sTNFR-1 and sTNFR-2 concentrations with kidney progression in patients with established kidney disease has been described. Notably, in the Multi-Ethnic Study of Atherosclerosis (MESA) conducted in patients without previous cardiovascular events and with preserved renal function (baseline eGFR > 60 mL/min/1.73 m^2^ in more than 90% of patients), sTNFR-1 concentrations were associated with faster decline in eGFR over the course of a decade, independent of previously known risk factors for kidney disease progression. Despite this association, a clear mechanism linking sTNFR-1 elevation and renal function decline has not been described [37]. Recently, using the AASK and VA-NEPHRON cohorts, it has been shown that among individuals with and without diabetes, longitudinal increases in sTNFR1 and sTNFR2 were associated with progressive CKD, independent of the initial biomarker levels and kidney function [38]. However, since the rate of progressive renal function decline in living donors is very low and the number of patients reaching end-stage renal disease is lower than 0.1%, post-donation increases in these biomarkers may have a different implication for living kidney donors.

Our study has some limitations related to the small sample size and the short follow-up, especially for living donors. In the donor cohort, the atherosclerotic burden was low and, due to the absence of CVE during follow-up, it was not possible to evaluate the impact of renal function decrease. Despite the changes in the levels of some biomarkers (sVCAM-1 and sTNFRs), renal function adaptation was fully preserved in the donor cohort until 5 years, suggesting that these changes may be linked to the decrease in renal function rather than acting as biomarkers of ED or subclinical inflammation.

## 4. Materials and Methods

### 4.1. Study Design

This is a non-interventional, prospective, multicenter, longitudinal, investigator-driven study (NCT02515643) involving two cohorts: living kidney donors and their transplant recipients. The participating centers were Hospital Universitari Vall d’Hebron and Hospital del Mar-Parc de Salut Mar in Barcelona, Hospital Universitario de Canarias in La Laguna-Tenerife, and Hospital Regional Universitario Carlos Haya in Malaga, Spain. The study protocol was approved by the Ethics Committee of the Hospital Universitari Vall d’Hebron (PR(AG)219-2014), and the donors and recipients signed an informed consent form before any study procedure. This study was conducted in accordance with the Declaration of Helsinki and was consistent with the Principles of the Declaration of Istanbul on Organ Trafficking and Transplant Tourism.

### 4.2. Patients

Inclusion and exclusion criteria applied in this study are based on the consensus document of the Spanish Nephrology Society and the National Organization of Transplantation to guide clinical practice in living renal transplantation [39,40].

Cohort 1: Healthy subjects who underwent nephrectomy as part of the living donor program at each participating center were invited to participate. Inclusion criteria were no history of familiar nephropathies and/other diseases that may increase the risk for future renal disease, age ≥ 18 years, isotopic or estimated GFR by the CKD-EPI formula > 80 mL/min/1.73 m^2^, microalbuminuria < 30 mg/g, normal urinary sediment, normal blood pressure or well-controlled hypertension with one antihypertensive drug without other risk factors for cardiovascular disease, no previous history of diabetes including gestational diabetes and fasting glucose < 126 mg/dL and 2 h serum glucose after a 75 g oral glucose tolerance test (OGTT) < 200 mg/dL. Exclusion criteria were history of cancer except non-melanoma skin cancer, history of vasculitis (e.g., lupus), sarcoidosis, gastrointestinal inflammatory diseases, autoimmune disease, history of major cardiovascular events, history of deep vein thrombosis or pulmonary embolism, active infection including hepatitis B, C and HIV infections, anatomic vascular variants precluding laparoscopic nephrectomy, renal stones except a solitary lithiasis < 1.5 cm once metabolic disorders were ruled out, major psychiatric disorders, active alcohol or drug abuse, obesity defined as body mass index > 35 kg/m^2^ and pregnancy. Left or right nephrectomy was performed with a laparoscopic technique by well-trained urologists at each participating center.

Cohort 2: This cohort consisted of recipients of renal transplants from cohort 1. The inclusion criteria were CKD stage 5 and a negative complement-dependent lymphocytotoxicity donor–recipient crossmatch. Exclusion criteria were glomerulonephritis with a high recurrence rate after transplantation, severe aorto-iliac atheromatosis precluding transplantation, major psychiatric disorders, alcohol and drug abuse, active infection, patients requiring desensitization treatment before transplantation and pregnancy.

### 4.3. Study Procedures

During the previous month before surgery and one year later, the following procedures were performed in donors and recipients:
Glomerular filtration rate before nephrectomy in donors and at one year in donors and recipients was measured by iohexol clearance (mGFR). The protocol for the study was guided by Instituto de Tecnologías Biomédicas, University of La Laguna (Tenerife, Spain) [41]. Iohexol levels in plasma samples obtained at each center were sent to this validated lab for their determination by HPLC [29]. Results of mGFR were not available to clinicians at the time of kidney donation. Additionally, the estimated glomerular filtration rate by CKD-EPI (eGFR-CKD-EPI) and MDRD-4 (eGFR-MDRD-4) using standard formulas was employed to estimate renal function in both cohorts.Urinary albumin to creatinine ratio (UACR) in an early-morning spot sample was determined locally by an immunoturbidimetric assay.An oral glucose tolerance test (OGTT) was performed at baseline and at 1 year in the donors. Insulin levels were also determined locally at both time periods to calculate the HOMA-IR.Serum samples for the measurement of endothelial dysfunction and low-grade inflammation markers were obtained and stored at each center. At the end of the study, all samples were sent to Hospital Universitari Vall d’Hebron laboratories (Barcelona) for their determination.In the donor cohort the following procedures were also performed:Ambulatory blood pressure monitoring (ABPM) with an overnight-automated monitor (Spacelab 90207; Spacelabs Healthcare, Snoqualmie, WA, USA) with appropriate cuff sizes for each patient was performed at baseline and at one year.Baseline atherosclerotic burden:A carotid ultrasound to determine the number of plaques and intima-media thickness (IMT) was performed in both carotid arteries with a high-frequency (8–12 MHz) linear transducer (ESAOTE, 7300, Florence, Italy). The numbers of carotid plaques in both arteries were added and the mean intima-media thickness (IMT) of both arteries was calculated.Carotid–femoral pulse wave velocity (PWV, m/s) was measured by pulse tonometry (Sphingmocor Atcor, EM3, Sidney, Australia).The ankle–brachial index (ABI) was determined by an automated blood pressure monitor with appropriate cuff sizes (Omron, Kyoto, Japan).

### 4.4. Treatments

All recipients received an immunosuppression schedule based on the combination of tacrolimus, mycophenolate mofetil, and corticosteroids. The use of induction therapy and steroid reduction/withdrawal were conducted as per local clinical practice to maintain recipients on a low dose of steroids from the third month onwards (5 mg/day). Modification of the immunosuppression schedule due to clinical indications (viral infection, cancer, or others) was performed according to local clinical practice. Patients with total serum cholesterol >200 mg/dL were treated with statins/ezetimibe according to local clinical practice. The target office blood pressure for both donors and recipients was <140/90 mmHg, and treatment was administered according to local clinical practice. The first-line anti-hypertensive drug for donors and recipients was a calcium channel blocker while, for patients with uncontrolled hypertension, an ACEi/ARB or thiazide was chosen as the second drug.

### 4.5. Biomarkers of Endothelial Dysfunction and Chronic Inflammation

Endothelial dysfunction: Circulating levels of soluble VCAM (vascular cell adhesion molecule), soluble ICAM (intercellular adhesion molecule), soluble E-selectin and PTX-3 (pentraxin) were determined by the microfluidics-based quantitative immunoassay, ELLA^®^ (Protein Simple, CA, USA) [42]. The serum concentration of PECAM (platelet/endothelial cell adhesion molecule) was determined by ELISA (Novus Biologicals, CO, USA). Determination of vWF (antigen of von Willebrand factor) serum levels was performed on an AcuStar instrument (Instrumentation Laboratories, Bedford, MA, USA), by using the HemosIL AcuStar VWF:Ag chemiluminescent cartridge reagent kit [43].Chronic inflammation: usPCR (ultrasensitive C-reactive protein) was determined by nephelometry. Circulating levels of IL-6 (interleukin 6), sTNFR1 and sTNFR2 (soluble tumor necrosis factor receptors 1 and 2) were determined using the microfluidics-based quantitative immunoassay, ELLA^®^ (Protein Simple, CA, USA) [42]. The serum concentration of sTWEAK (soluble TNF-like weak inducer of apoptosis) was determined by ELISA (DuoSet, Minneapolis, MN, USA) [44].

### 4.6. Data Analysis

Results were expressed as frequencies for qualitative variables, as mean ± standard deviation for variables with normal distribution and as median and interquartile range for non-normally distributed variables. Normal distribution was evaluated by the Shapiro–Wilk test. Analysis of paired data was performed using appropriate statistics (the chi-squared test for categorical variables, the Student *t*-test for paired data for continuous variables with normal distribution and the Wilcoxon test for ordinal or continuous variables not normally distributed). Analysis of unpaired data was performed using appropriate statistics (the chi-squared test for categorical variables, the Student *t*-test for unpaired continuous variables with normal distribution and the Mann–Whitney U test for ordinal or continuous variables not normally distributed). Lineal regression analysis was used to evaluate associations between quantitative variables and Spearman correlation for non-parametric data. Correlations between variables were considered negligible (r < 0.2), weak (r from 0.2 to 0.39) or moderate (r from 0.4 to 0.6).

Bland–Altmananalysis was employed to analyze the relationship between mGFR and eGFR-CKD-EPI and mGFR and eGFR-MDRD-4 at different time points. A linear mixed-effects model with random intercept was employed to analyze the evolution of renal function from the first month to 5 years in both cohorts.

All *p*-values were two-tailed and a *p*-value < 0.05 was considered significant.

## 5. Conclusions

In this prospective study, we observed an increase in the levels of some ED and subclinical inflammation biomarkers (e.g., sVCAM-1 and sTNFR1/sTNFR2) in living kidney donors and a decrease in their renal transplant recipients. These findings agree with the widely reported reduced CV risk after renal transplantation in ESRD patients. However, these changes were not associated with a modification of CV risk or CKD progression until five years after nephrectomy in the donor cohort. Thus, the usefulness of these biomarkers for assessing CV risk in living kidney donors deserves further studies with a longer follow-up period.

## Figures and Tables

**Figure 1 ijms-26-09535-f001:**
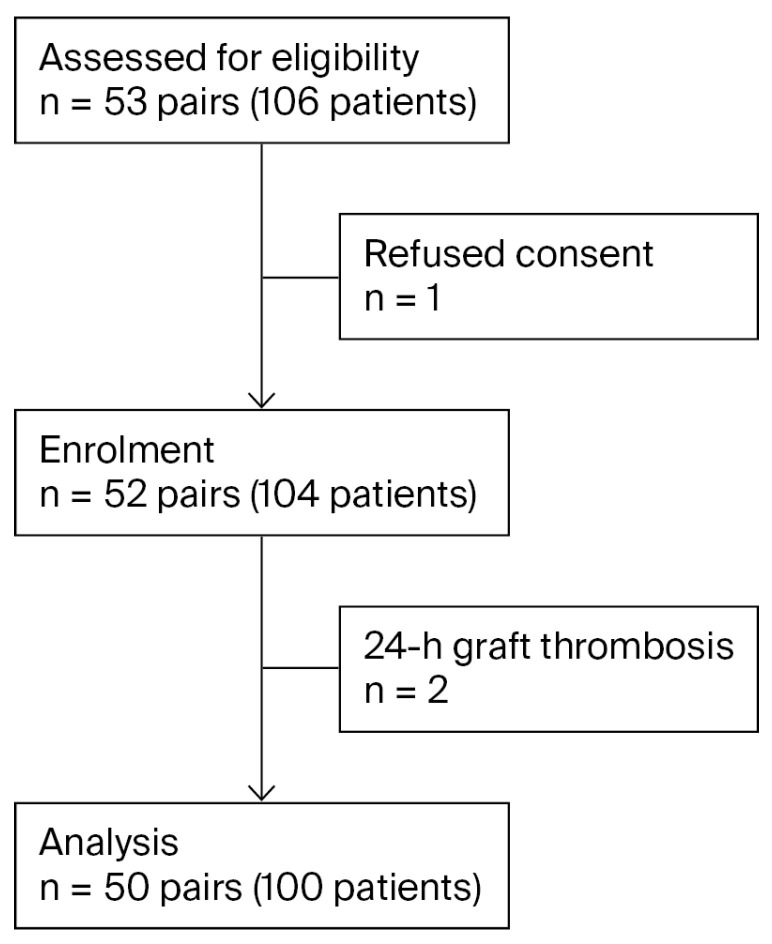
CONSORT diagram.

**Figure 2 ijms-26-09535-f002:**
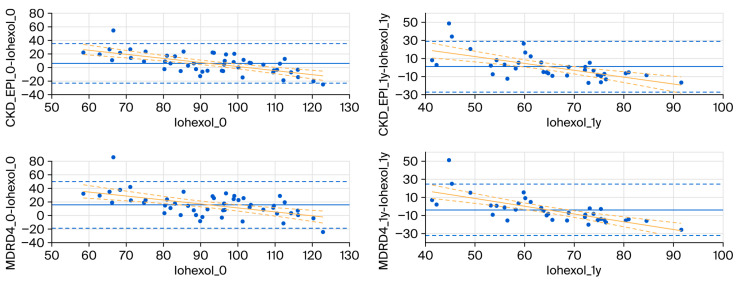
Bland–Altman plots of donor baseline (left panel) and 1-year (right panel) GFR measured by iohexol clearance and estimated GFR by CKK-EPI and MDRD-4 formulas. Regression lines with 95% confidence interval are shown.

**Figure 3 ijms-26-09535-f003:**
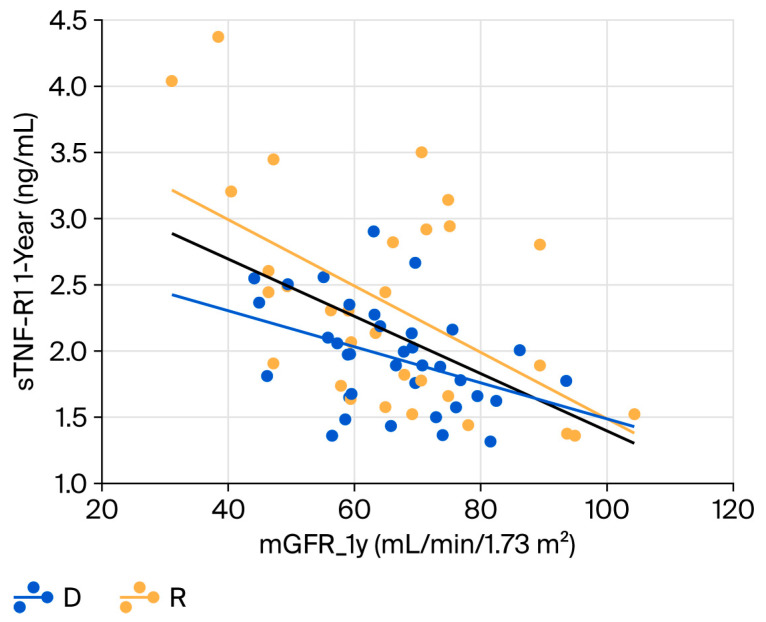
Regression analysis between the measured glomerular filtration rate and sTNFR1 at 1-year follow-up in the donors and recipients. R = 0.49 and *p* < 0.001 for all patients; R = 0.40 and *p*-value = 0.016 for the donor cohort; and R = 0.55 and *p*-value = 0.001 for the recipient cohort. The intercept is different between the two cohorts (4.00 for donors and 2.85 for recipients, *p*-value = 0.004), but the slope is not significantly different (−0.025 for donors and −0.012 for recipients, *p*-value = 0.233).

**Figure 4 ijms-26-09535-f004:**
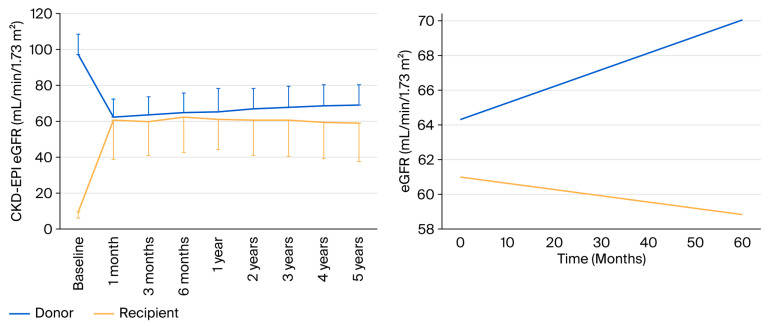
Estimated GFR according to the CKD-EPI formula from baseline to 5 years in the donor and recipient cohort, expressed as mean ± SD (**left panel**). Model-based adjusted means of eGFR (mL/min/1.73 m^2^) from the first month after donation/transplantation to 5 years, estimated with a linear mixed-effects model with random intercept by patient cohort (**right panel**).

**Figure 5 ijms-26-09535-f005:**
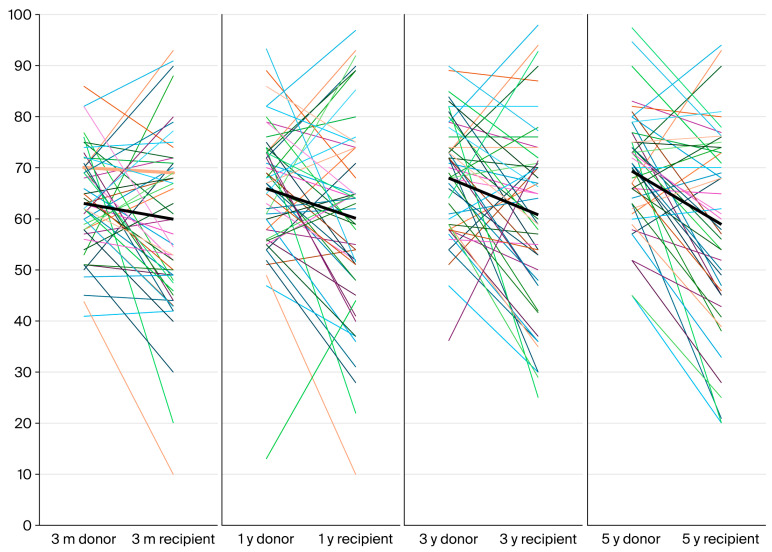
Paired donor–recipient eGFR at 3 months, 1 year, 3 years, and 5 years in both cohorts. The mean eGFR for each cohort is shown as a thick black line.

**Table 1 ijms-26-09535-t001:** Demographic and anthropometric data and pre-transplant comorbidities in donor and recipient pairs.

Variable	Donors	Recipients
Age (years)	52 ± 12	48 ± 14
Sex (male/female)	20/30	29/21
Race (Caucasian/Hispanic)	46/4	47/3
Height (cm)	165 ± 9	168 ± 10
Weight (kg)	72 ± 13	74 ± 16
BMI (kg/m^2^)	26.0 ± 3.8	26.2 ± 5.2
Hypertension (no/yes)	42/8	7/43
Diabetes (no/yes)	50/0	44/6
Smoker (never/past/current)	32/8/10	30/14/6
Office systolic blood pressure (mm Hg)	126.4 ± 14.8	137.6 ± 17.9
Office diastolic blood pressure (mm Hg)	73.2 ± 9.4	83.1 ± 12.4
Mean office blood pressure (mm Hg)	91.0 ± 10.0	101.3 ± 13.0
Serum glucose (mg/dL)	91 ± 6	125 ± 42
Total cholesterol (mg/dL)	219 ± 41	194 ± 38
LDL cholesterol (mg/dL)	140 ± 36	125 ± 35
Triglycerides (mg/dL)	113 ± 37	109 ± 68
Cause of ESRD		
GN/CTIN/ADPKD/DN/vascular/others		12/9/7/3/5/14
Pre-emptive/HD/PD		30/15/5
Time on RRT (mo.)		11 ± 4

BMI, body mass index; ESRD, end-stage renal disease; GN, chronic glomerulonephritis; CTIN, chronic tubule-interstitial nephritis; ADPKD, autosomal dominant polycystic kidney disease; DN, diabetic nephropathy; HD, hemodialysis; PD, peritoneal dialysis; Time on RRT, time on renal replacement therapy; mo., months.

**Table 2 ijms-26-09535-t002:** Renal function in donor–recipient pairs at baseline and 1-year follow-up.

	Donors		Recipients	
	Baseline	1 Year	Baseline	1 Year
**Creatinine (mg/dL)**	0.75 ± 0.15	1.08 ± 0.22	6.01 ± 2.62	1.36 ± 0.41
**eGFR CKD-EPI**	98 ± 13	66 ± 11	10 ± 4	64 ± 17
**eGFR MDRD-4**	95 ± 16	61 ± 10	10 ± 4	56 ± 15
**mGFR**	93 ± 17	65 ± 12	n.a.	57 ± 13

eGFR, estimated glomerular filtration rate (mL/min/1.73 m^2^); CKD-EPI, chronic kidney disease epidemiology collaboration formula; MDRD-4, modification diet of renal disease formula; mGFR, measured glomerular filtration rate by iohexol clearance (mL/min/1.73 m^2^); n.a., not applicable.

**Table 3 ijms-26-09535-t003:** Levels of endothelial dysfunction and inflammation biomarkers in donor–recipient pairs at baseline and 1-year follow-up.

	Donors		Recipients	
	Baseline	1 Year	Baseline	1 Year
**VCAM-1 [ng/mL]**	635 ± 197	718 ± 213 *	1208 ± 417	943 ± 313 *
**ICAM-1 [ng/mL]**	382 ± 91	407 ± 101	430± 159	433 ± 109
**E-selectin [ng/mL]**	29 ± 10	30 ± 11	37 ± 19	34 ± 14 **
**PECAM-1 [ng/mL]**	73 ± 13	80 ± 14	79 ± 23	71 ±15 *
**vWF [%]**	96 ± 36	103 ± 35	181 ± 53	167 ± 57
**PTX-3 [ng/mL]**	2.7 ± 1.8	3.4 ± 2.1	3.7 ± 3.1	3.0 ± 2.3
**UACR [mg/g]**	5.9 ± 6.3	10.5 ± 24.0	-	-
**IL-6 [pg/mL]**	4.6 ± 5.7	6.1 ± 10.8	7.7 ± 9.0	6.1 ± 3.6
**TNFR1 [ng/mL]**	1.4 ± 0.9	1.9 ± 0.4 *	10.2 ± 5.6	2.4 ± 0.8 *
**TNFR2 [ng/mL]**	2.8 ± 1.2	3.8 ± 0.7 *	12.2 ± 4.0	4.8 ± 1.9 *
**TWEAK [pg/mL]**	544 ± 499	493 ± 100	437 ± 107	465 ± 122
**hsCRP [mg/dL]**	0.36 ± 1.04	0.31 ± 0.39	0.41 ± 0.69	0.36 ± 0.41

VCAM-1, soluble VCAM (vascular cell adhesion molecule); ICAM-1, soluble ICAM (intercellular adhesion molecule); PECAM-1, soluble platelet/endothelial cell adhesion molecule; vWF, von Willebrand factor; PTX-3, pentraxin 3; UACR, urinary albumin to creatinine ratio; IL-6, interleukin 6; TNFR1, soluble tumor necrosis factor receptor-1; TNFR2, soluble tumor necrosis factor receptor-2; TWEAK, soluble TNF-like weak inducer of apoptosis; hsCRP, high-sensitivity C-reactive protein. * *p* < 0.01; ** *p* < 0.05 All comparisons were made via paired *t*-test.

**Table 4 ijms-26-09535-t004:** Ambulatory blood pressure monitoring in the donor cohort at baseline and 1-year follow up.

Time Point	Office BP		ABPM-Day		ABPM-Night		ABPM-Pattern
	SBP	DBP	SBP	DPB	SBP	DPB	Non-Dipper (%)
Baseline	124 ± 15	76 ± 9	122 ± 13	76 ± 9	109 ± 11	66 ± 8	18%
1 year	129 ± 15	77 ± 10	120 ± 10	77 ± 8	108 ± 9	66 ± 8	34%

## Data Availability

Data from the present study can be made available upon reasonable request.

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
