# Peer review of "Estimating the Contribution of Renal Function to Endothelial Dysfunction and Subclinical Inflammation with a Two-Cohort Study: Living Kidney Donors and Their Transplant Recipients"

_ijms, 2025, doi:10.3390/ijms26199535_

Round 1
Reviewer 1 Report
Comments and Suggestions for Authors
In the manuscript “Estimating the contribution of renal function to endothelial dysfunction and sub-clinical inflammation by a two-cohort study: living kidney donors and their transplant recipients”, the authors employed fifty pairs of living donors and the corresponding recipients. They found that sVCAM-1 among endothelial dysfunction biomarkers increased in donors and decreased in recipients while sTNFR1 and sTNFR2 among inflammation biomarkers significantly increased in donors and decreased in recipients. After transplantation, a parallel impairment and improvement in ED and subclinical inflammation biomarkers was observed in the donor and recipient cohort, respectively. This work is valuable for clinical evaluation for livinbf kidney transplantation.
1, The authors employed their objectives between 2015-2018 and followed 1 year. Now it is already 2025, therefore, I suggest the authors would provide the date from 3-5 year follow-up.
2, The authors mentioned cardiovascular risk in this work. They observed the related indexes such as BMI, blood pressure, total cholesterol, LDL and triglycerides. Do they also have the indexes from images or/and ultrasound about cardiovascular status?
3, Please describe the changes of GFR/Cr in each pair of donor and the corresponding recipient post transplantation at each timepoint such as day 0, 1m,2m,3m,6m,9m,1y,2y,3y,5y.
Author Response
"Please see the attachment."

Reviewer 2 Report
Comments and Suggestions for Authors
Torres et al submit a research article entitled "Estimating the contribution of renal function to endothelialdysfunction and sub-clinical inflammation by a two-cohort study: living kidney donors and their transplant recipients".
The authors studied renal function, endothelial dysfunction (ED) and subclinical inflammation biomarkers in two cohorts of patients by a non-interventional, prospective, multicenter, longitudinal study. Overall, the article is interesting, with a comparison between donors and recipietns in a spanish population. This is really relevant in the CKD field as a possible link between GFR reduction in living donors and CV risk is controversial.
Among endothelial dysfunction biomarkers, sVCAM-1 increased in donors and decreased in recipients while amongst inflammation biomarkers, sTNFR1 and sTNFR2 significantly increased in donors and decreased in recipients. There already were links described between CVD risk and endothelial dysfunction biomarkers in CKD patients (PMID: 34638892). Perhaps the authors can discuss this ?
In Table 1, I am surprised that the authors did not check statistical differences between donors and recipients.
The study protocol was approved by the Ethics Committee at each participating
center: please give specific numbers for each authorization.
I would suggest the authors consider innovative ways to assess kidney rejection to enrich the discussion. For example, other authors have measured the plasma levels of extravesicles in kidney transplant patients. They found that patinets with antibody-mediated rejections had low plasma levels of EVs. (see PMID: 40332150).
The paper has quite a lot of grammar approximation, and should be corrected by an english speaking colleague.
A final recapitulative figure would be welcome for the reader.
Minor
in abstract, "two cohorts" is repeated twice in the same sentence.
please correct " chronic kidney disease (CKD) has become a worldwide threaten" in intro
change "Healthy subjects who undergone nephrectomy as part of the living donor" to "Healthy subjects who underWENT nephrectomy as part of the living donor"
line 268 " Additionally, there existS a proportional bias"
Comments on the Quality of English LanguageThe paper has quite a lot of grammar approximation, and should be corrected by an english speaking colleague.
Author Response
"Please see the attachment."

Reviewer 3 Report
Comments and Suggestions for Authors
I read with interest the manuscript entitled "Estimating the contribution of renal function to endothelial dysfunction and subclinical inflammation by a two-cohort study: living kidney donors and their transplant recipients".
This approach is valuable because it leverages a natural experimental setup—donors as a baseline with good renal function and recipients with compromised function—to explore causative links.
The abstract needs to be more structured in relation to the content of the study, especially the results.
The introduction is well-designed with a clear aim at the end.
Methodologically, rigorous participant selection, standardized biomarker assessment, appropriate control groups, and thorough data analysis adjusting for key confounders are essential.
Inclusion and exclusion criteria are clearly stated.
When you state "local practice", please provide a more detailed explanation for the sake of transparency and reproducibility of the study.
For the determination of biomarkers of endothelial dysfunction and chronic inflammation, please indicate the used protocols described in previous studies with a reference in parentheses.
What test did you use to test the normality of the data distribution?
Is the sample size adequate to detect significant differences, considering the variability of the biomarkers? Please explain.
Use of multivariate analyses to adjust for potential confounders.
In tables, list p-values in a separate column with superscripts indicating which test was used in the calculation.
The boxplots next to the tables are unnecessary. Please remove them.
Be careful when using the terms "Relationship", "association". Think of correlation! Also, for the interpretation of the results, you need to methodologically describe what constitutes excellent, good and poor correlation in your study. Give intervals to interpret the results transparently.
At the beginning of the discussion, please present the most relevant results of your study.
Many common facts are woven throughout the discussion, which are more appropriate for an introduction. Focus the discussion strictly on comparing your results with similar studies on the topic, of which there is no shortage.
You have not touched on many of the results in your discussion. Please touch on all of them with critical evaluation. Also, create a section on limitations that you may not be aware of. Think about it.
Please emphasize in your conclusion what your study brings new in relation to the existing literature. I assume that your conclusion is generally generated. It must be specific.
The references are not written according to the instructions for authors. Please correct them.
Considering that the study was registered more than 10 years ago, and the study itself was conducted more than 7 years ago, why are you only planning to publish the article now?
Author Response
"Please see the attachment."

Round 2
Reviewer 1 Report
Comments and Suggestions for Authors
The revised version has been modified as my suggestion. I have no more comments.
Author Response
Thank you very much for helping us to improve the manuscript.
Reviewer 2 Report
Comments and Suggestions for Authors
Changes are ok
Author Response

(The authors gave the same response as above.)

Reviewer 3 Report
Comments and Suggestions for Authors
Thank you for your answers and explanations.
I ask you to adopt all the previously mentioned suggestions.
Author Response
Thank you very much for helping us to improve our manuscript.